# Pathophysiological Effects of Various Interleukins on Primary Cell Types in Common Heart Disease

**DOI:** 10.3390/ijms24076497

**Published:** 2023-03-30

**Authors:** Yong Liu, Donghui Zhang, Dan Yin

**Affiliations:** 1State Key Laboratory of Biocatalysis and Enzyme Engineering, School of Life Science, Hubei University, Wuhan 430062, China; 2Hubei Province Key Laboratory of Biotechnology of Chinese Traditional Medicine, National & Local Joint Engineering Research Center of High-Throughput Drug Screening Technology, Hubei University, Wuhan 430062, China

**Keywords:** heart disease, interleukin, cardiomyocytes, fibroblasts, immune cells

## Abstract

Myocardial infarction (MI), heart failure, cardiomyopathy, myocarditis, and myocardial ischemia-reperfusion injury (I/R) are the most common heart diseases, yet there is currently no effective therapy due to their complex pathogenesis. Cardiomyocytes (CMs), fibroblasts (FBs), endothelial cells (ECs), and immune cells are the primary cell types involved in heart disorders, and, thus, targeting a specific cell type for the treatment of heart disease may be more effective. The same interleukin may have various effects on different kinds of cell types in heart disease, yet the exact role of interleukins and their pathophysiological pathways on primary cell types remain largely unexplored. This review will focus on the pathophysiological effects of various interleukins including the IL-1 family (IL-1, IL-18, IL-33, IL-37), IL-2, IL-4, the IL-6 family (IL-6 and IL-11), IL-8, IL-10, IL-17 on primary cell types in common heart disease, which may contribute to the more precise and effective treatment of heart disease.

## 1. Introduction

Heart disease is the leading cause of death worldwide [1]. Heart disease generally includes myocardial infarction (MI), heart failure, cardiomyopathy, myocarditis, and myocardial ischemia-reperfusion injury (I/R) [2]. There are no effective therapy methods currently because of their complex pathogenesis. Heart disease is characterized by the involvement of four major cell types: cardiomyocytes (CMs), fibroblasts (FBs), endothelial cells (ECs), and immune cells [3]. CMs, FBs, and ECs are resident cells in the heart; however, immune cells are infiltrated during inflammation, and they play a significant role in the pathophysiological process of heart disease. Thus, targeting a specific cell type for the treatment of heart disease may be more effective.

Inflammation has been identified as a potential initiator and promoter of heart disease. Different interleukin (IL) families have been linked to the development of heart disease, either positively or negatively. Proinflammatory factors such as IL-1β antagonists or antibodies have shown promising results in early clinical trials for heart disease [2]. Despite this, researchers are still exploring more effective cytokine treatment methods. However, the same interleukin may have various effects on different cell types of the heart. For instance, IL-11 has been found to mediate cytoprotective signals in cardiomyocytes [4], yet it has also been reported to have a profibrotic effect in cardiac fibrosis [5].

The IL-1 family is a major cytokine family associated with various cardiovascular diseases, initially comprising only two forms, IL-1α and IL-1β. To date, the IL-1 family has expanded to include 11 members (IL-1α, IL-1β, IL-1ra, IL-18, IL-33, IL-36Ra, IL-36α, IL-36β, IL-36γ, IL-37, and IL-38), with most of them being proinflammatory, while IL-1ra, IL-36Ra, IL-37, and IL-38 are anti-inflammatory [6].

IL-2 is an O-glycosylated four alpha-helix bundle cytokine that is primarily produced by activated T cells, dendritic cells, and B cells. It plays a pivotal role in the immune response to heart disease by regulating B-cell proliferation and immunoglobulin production, as well as maintaining T-cell homeostasis [7].

IL-4 is a glycosylated, type I cytokine with three intrachain disulfide bridges. It is mainly produced by T cells, natural killer T cells, mast cells, and eosinophils. In addition, it plays a central dual role in the development of inflammation in heart disease [8].

IL-6 is the founding member of the IL-6 cytokine family, which also includes IL-11, IL-27, IL-30, IL-31, leukemia inhibitory factor (LIF), oncostatin M (OSM), cardiotrophin-like cytokine (CLC), ciliary neurotrophic factor (CNTF), cardiotrophin-1 (CT-1), and neuropoietin. It is produced by numerous different cell types and is essential for regulating heart disease progression. IL-6-IL-6 R alpha complex promotes gp130 dimerization and the formation of a heterohexameric complex [9].

The CXCL8 gene encodes interleukin-8 (IL-8), a key mediator with both deleterious and beneficial properties. Multiple studies have reported elevated levels of IL-8 in various cardiac pathologies, including MI, suggesting that IL-8 could be a potential therapeutic target for heart disease [10].

The IL-10 family is composed of six members, namely IL-10, IL-19, IL-20, IL-22, IL-24, and IL-26. As the founding member, IL-10 has anti-inflammatory and immunosuppressive properties that serve to prevent excessive inflammation. It has been demonstrated to inhibit the antigen presentation capabilities of monocytes, macrophages, and dendritic cells, while simultaneously enhancing their tolerance-inducing, scavenger, and phagocytic functions. Additionally, IL-10 has been shown to suppress Th1-, Th2-, and Th17-mediated immune responses by inhibiting the proliferation of CD4+ T cells and their ability to produce proinflammatory cytokines. Furthermore, it has been observed to inhibit the secretion of proinflammatory mediators by neutrophils, eosinophils, and mast cells, as well as mast cell development.

IL-17 has been demonstrated to exert a wide range of biological activities on a variety of cell types, including CMs, ECs, neutrophils, monocytes, and macrophages. This cytokine has been shown to be involved in the inflammatory response in heart disease, suggesting a potential role in the pathogenesis of this condition [11].

Collectively, these findings suggest that IL plays a significant role in the pathogenesis of heart disease, and further research is needed to elucidate the precise mechanisms by which it modulates primary cell types in the heart. We have searched for papers using PubMed and Google Scholar with keywords: interleukins, IL-1, IL-18, IL-33, IL-37, IL-2, IL-4, IL-6, IL-11, IL-8, IL-10, IL-17, and heart disease. Then, we selected approximately 120 papers that involved the pathophysiological role of interleukins on primary cell types in heart disease as a reference. This review will primarily focus on the pathophysiological effects of various interleukins including the IL-1 family (IL-1, IL-18, IL-33, IL-37), IL-2, IL-4, the IL-6 family (IL-6 and IL-11), IL-8, IL-10, IL-17 on primary cells of heart disease-CMs, FBs, ECs, and immune cells. It would be more effective to focus on a certain cell type for the therapy of heart disease.

## 2. Cardiomyocytes (CMs)

CMs are the beating muscle cells that make up the atria and ventricles and are being targeted primarily in heart disease therapy. The specific effects of different interleukins on CMs in common heart disease are listed in Figure 1.

### 2.1. IL-1

IL-1α and IL-1β are proinflammatory cytokines and their levels are correlated with the severity and pathogenesis of heart disease. Targeting the IL-1 signaling cascade including IL-1α and IL-1β may be a promising therapeutic target for patients with MI [12,13]. Moreover, in vivo MI mouse models have shown that inhibition of IL-1α reduces myocardial I/R damage, resulting in the retention of left ventricular function, reduced infarction area, and decreased activation of inflammatory bodies [14]. Thus, IL-1α blockers may represent an effective therapeutic approach to reduce I/R damage to the heart.

In a mouse model of MI, dead cardiomyocytes will release IL-1α [15,16]. In addition, the release of IL-1β in fulminant myocarditis leads to extensive inflammation, leading to the further death of cardiomyocytes, the gradual loss of active contracted tissue, and the development of cardiomyopathy and HF.

IL-1β is concentration-dependent and may elevate myocardial ring GMP through the myocardial L-arginine-NO pathway, leading to the restriction of systolic ejection and cardiac depression [13]. The regulation of the excitatory-contractile coupling of cardiomyocytes is reflected in changes in contractile force, and cytokine-specific effects appear to exist in the excitatory–contraction coupling, with TNF-α and IL-1β affecting inward calcium currents [17,18]. IL-1β has been shown to significantly prolong the duration of the action potential of guinea pig ventricular cells by changing the conductance of calcium channels [19]. IL-1β has been shown to rapidly inhibit the voltage-dependent Ca^2+^ current in adult rat ventricular muscle cells. Consistent with these data, IL-1β has been shown to inhibit systolic cardiomyocyte function by potentially involving the destruction of calcium processing or the inhibition of a β-adrenergic response [20,21]. In patients with HF, IL-1β has been demonstrated to decrease the beta-adrenergic responsiveness of L-type calcium channels, as well as decrease calcium homeostasis genes, including phospholamban and sarcoplasmic reticulum calcium ATPase [22,23]. Furthermore, IL-1β has been shown to have a proapoptotic effect on cardiomyocytes [24] and exert negative inspiratory effects on both isolated cardiomyocytes and intact hearts [13,25].

### 2.2. IL-2

Recent studies have demonstrated that high doses of IL-2 may induce AMI [26]. In isolated normal myocytes, IL-2 was found to decrease the amplitude of calcium transients induced by electrical stimulation, likely through blocking Ca^2+^ ATPase activity in the sarcoplasmic reticulum [27]. Furthermore, IL-2 concentrations produced by CD4+ T lymphocytes were abnormally elevated in patients with DCM, which may reflect deficiencies in T-cell function in these patients [28].

However, there are also studies demonstrating the protective role of IL-2 in MI. The injection of IL-2-activated NK cells has been shown to promote vascular remodeling through a4b7 integrin and killer cell lectin-like receptor (KLRG)-1 and promote cardiac repair after MI [29,30]. In addition, Cao et al. have reported that IL-2 could reduce infarct size by activating kappa-opioid receptors [31]. Moreover, IL-2 can be stimulated by the IL-2IgG2b fusion protein to improve left ventricular (LV) contraction function and remodeling in an MI rat model [30].

In conclusion, additional experimental studies are needed to fully elucidate the role of IL-2 and develop its therapeutic potential in heart disease.

### 2.3. IL-4

IL-4 is generally regarded as an anti-inflammatory cytokine. A recent study by Wan et al. showed that Vγ1+ γδT cells, as one of the main early producers of IL-4 after acute viral infection, protect the mouse heart from acute viral myocarditis. Moreover, the neutralization of IL-4 in mice led to exacerbations of acute myocarditis, confirming the IL-4-mediated Vγ1 protective mechanism [32]. This finding was further supported by another study on viral myocarditis, which showed elevated levels of IL-4 in mice with attenuated viral myocarditis and elevated levels of heart expression [33].

However, a contradictory finding was reported in the context of autoimmune myocarditis, where eosinophils are the predominant cell type in the heart expressing IL-4, and eosinophil-specific IL-4 deletion leads to improved cardiac function. In this regard, eosinophils have been shown to drive myocarditis progression to inflammatory dilated cardiomyopathy (DCMi), and this process is mediated by IL-4 [34].

### 2.4. IL-6

IL-6, as an upstream marker of inflammation, is independently associated with the risk of major adverse cardiovascular disease events, MI, HF, and cancer mortality stable coronary heart disease [35]. IL-6 and sIL-6R have been associated with AMI and cardiac injury; binding to trans-IL-6 receptors alters intracellular signaling, and blocking IL-6 receptor binding may be a causative factor in AMIs [36]. Hypothetically validated, the IL-6 receptor antagonist tocilizumab reduces inflammation and the release of TnT in non-ST-segment elevation MI (NSTEMI). Therefore, IL-6 is a potential therapeutic target for MI [37].

Cytokine-specific action appears to be present in the excitatory–contraction coupling and TNF-α IL-6 modulates Ca^2+^ ATPase activity of the sarcoplasmic reticulum in cardiomyocytes [38]. It was reported that the degradation of IL-6 mRNA inhibits the proinflammatory action in the stress-overloaded myocardium [39]. Moreover, the gene deletion of IL-6 improves cardiac function and weakens hypertrophy by eliminating the dependent effects of CaMKII on cardiomyocytes in the stress-overloaded myocardium [40]. In addition, in the model of LV remodeling after MI, a drug blockade of IL-6 through the administration of an anti-IL6R antibody weakens dilation and improves contraction function [41]. IL-6 can indirectly enhance the expression of iNOS, and excess nitric oxide may reduce myocardial contractility and may have toxic effects by triggering apoptosis [42].

IL-6 exerts negative inotropic action [43] and promotes a hypertrophy response in cardiomyocytes [44,45,46] through the gp130/STAT3 pathway, but can also enable protective action, mediated by mitochondrial function preservation [47]. Recombinant IL-6 induces a cytoprotective effect to prevent I/R damage and activates ERK1/2, JNK1/2, p38-MAPK, and PI3K without inducing STAT1/3 phosphorylation. These data suggest that the cardiomyocyte protective effect of IL-6 in I/R occurs through ERK1/2 and PI3K activation, but is not related to sIL-6R and JAK/STAT signaling [48].

### 2.5. IL-8

IL-8 is important in the development of MI. Serum IL-8 concentrations show a transient increase in the very early stages of AMI [49]. Specific monoclonal antibodies that neutralize IL-8 significantly reduce the degree of necrosis in rabbit myocardial I/R injury models [50]. High levels of IL-8 in STEMI patients with HF are associated with less improvement in left ventricular function in the first 6 weeks after PCI, suggesting that IL-8 may play a role in reperfusion-related injuries to the myocardium after ischemia [51].

### 2.6. IL-10

Cardioprotective effects of IL-10 on the cardiomyocytes of heart diseases were found in a variety of previous studies. The involvement of the Akt and Jak/Stat pathways in regulating TNF-induced cardiomyocyte apoptosis by IL-10 has been studied [52]. Subsequent research revealed that IL-10’s negative control of TNF-induced apoptosis was mediated by Akt via STAT3 activation [53]. In addition, there is a study revealing that IL-10-induced antiapoptotic signaling in cardiomyocytes includes upregulating TLR4 through MyD88 activation [54]. Moreover, Kishore, R et al. found that IL-10 attenuates pressure overload-induced hypertrophic remodeling and improves heart function via STAT3-dependent inhibition of NF-κB [55]. Exercise reduces HFD-induced cardiomyopathy by reducing obesity, inducing IL-10, and reducing TNF-α [56].

### 2.7. IL-11

IL-11 mediates cytoprotective signals in cardiomyocytes by activating phosphorylated STAT3 translocating into nuclei [57]. IL-11 attenuated cardiac remodeling after MI through the gp130/STAT3 axis [5]. In addition, it also reduced the I/R injury through STAT3 activation in the hearts [58]. This evidence demonstrates the therapeutic role of IL-11 in heart disease.

### 2.8. IL-17

As a proinflammatory cytokine, IL-17 participates in an array of heart diseases. IL-17 was reported to have contributed to the process of cardiac fibrosis, the activation of matrix metalloproteinases, and enhanced cardiac cell death. IL-17 induces mouse cardiomyocyte apoptosis via Stat3-iNOS activation, suggesting that IL-17 contributes to cardiac damage [59]. It has also been observed that IL-17A induces cardiomyocyte apoptosis through the p38 mitogen-activated protein kinase (MAPK)-p53-Bax signaling pathway and promotes both early- and late-phase post-MI ventricular remodeling [60]. Pan et al. found that IL-17 affects the calcium-handling process involved in HF. The treatment of neonatal cardiomyocytes with steady-state concentrations of IL-17 suppressed transient calcium and decreased SERCA2a and Ca_v_1.2 expression, this effect is mediated via the NF-κB pathway [61].

### 2.9. IL-18

IL-18 is a proinflammatory cytokine produced during various heart diseases. IL-18 was discovered to be elevated in animal models of AMI, HF, pressure overload, and LPS-induced dysfunction. Furthermore, IL-18 has been shown to regulate cardiomyocyte hypertrophy, induce cardiac systolic dysfunction, and lead to extracellular matrix remodeling [62,63]. Several observations suggest that IL-18BP is a potential therapeutic tool for reducing myocardial dysfunction caused by ischemia [64,65,66].

The treatment of HL-1 cardiomyocytes with IL-18 resulted in hypertrophy and elevated levels of ANP, likely via the activation of signaling pathways involving PI3K, Akt, and the transcription factor GATA4 [67]. In vitro studies found that the IL-18 treatment of cardiomyocytes increased peak and diastolic calcium transients and decreased the shortening of isolated cardiomyocytes [68]. Moreover, an increase in serum IL-18 concentration may induce apoptosis in cardiomyocytes, leading to ongoing myocardial injury in acute MI [69].

However, there are some controversial studies; for example, the expression of IL-18RNA in the myocardium of patients with dilated cardiomyopathy is downregulated [70], and IL-18 has been shown to play a beneficial role in viral myocarditis caused by the cerebrocarditis virus. The systemic administration of IL-18 is beneficial in mice with myocarditis and may be mediated by reducing the expression of TNF-α in the heart [65].

Overall, the role of IL-18 in heart disease is primarily in amplifying myocardial dysfunction, and the level also was recognized as a marker of heart injury in patients.

### 2.10. IL-33

IL-33 belongs to the IL-1 family. IL-33 and its receptor ST2 (located on the membrane of CMs) were demonstrated to be cardioprotective. The highly localized signaling pathway mediated by ST2 regulates the heart’s response to pressure overload. It was suggested that IL-33 secretion by endothelial cells is crucial in converting myocardial pressure overload into a selective systemic inflammatory state [71]. In a study involving wild-type mice, treatment with recombinant IL-33 was found to reduce angiotensin II and phenylephrine-induced cardiomyocyte hypertrophy and fibrosis. Furthermore, IL-33 treatment improved survival after transverse aortic constriction (TAC), a surgical procedure used to induce cardiac hypertrophy and heart failure in animal models [72].

### 2.11. IL-37

IL-37, like IL-10, is an anti-inflammatory interleukin generated by a variety of cell types. IL-37 is the main cytokine in the regulation of immune response, mainly inhibits the expression, production, and effect of proinflammatory cytokines, and plays a role in autoimmune diseases and organ transplantation [73]. The expression level of IL-37 is known to be low under normal physiological conditions; however, the expression level of IL-37 is significantly upregulated in response to an inflammatory environment, such as in patients with acute myocardial infarction (AMI) [74].

IL-37 plays an active role in a variety of cardiovascular diseases [75]. The Zeng group reported that human recombinant IL-37 can inhibit neutrophil infiltration and reduce cardiomyocyte apoptosis through a tail vein injection into myocardial I/R mice, thereby alleviating myocardial I/R injury in mice [76]. The team also reported that the intraperitoneal injection of human recombinant IL-37 and the intravenous injection of IL-37 and troponin co-induced dendritic cells can alleviate adverse ventricular remodeling after MI and cardiomyocyte apoptosis in mice, also attenuating the degree of cardiac fibrosis [77]. Overall, the positive role of IL-37 on other heart diseases, i.e., HF, needs to be further elucidated.

Overall, the effects of interleukins on CMs in heart disease are complex and context-dependent, and more research is needed to fully understand their roles in these conditions.

## 3. Fibroblasts (FBs)

Fibroblasts are the most abundant cell type found in connective tissue and are responsible for secreting collagen proteins that provide a structural framework for many tissues. Additionally, fibroblasts play an important role in wound healing. In heart disease, however, fibroblasts are transdifferentiated into activated myofibroblasts, which express α-smooth muscle actin (ACTA2) and secrete extracellular matrix (ECM) proteins and are a defining feature of fibrosis. The specific effects of different interleukins on FBs in common heart disease are summarized in Figure 2.

### 3.1. IL-1

In fibroblasts, IL-1 can induce the expression of extracellular matrix proteins, including collagen, and promote fibroblast proliferation and differentiation into myofibroblasts. IL-1 promotes matrix degradation phenotypes in fibroblasts in an IL-1R1-dependent manner and helps disrupt key stromal–cardiomyocyte interactions needed for cell survival in the infarcted myocardium [78]. IL-1-driven matrix degradation may eventually activate fibroblast-mediated matrix protein synthesis, leading to increased fibrosis by increasing the expression of fibroblast growth factor [79].

### 3.2. IL-4

IL-4 stimulates the inflammatory response, activates collagen synthesis, promotes fibrosis progression, and inhibits the production of anti-inflammatory cytokines. Urine IL-4 is associated with myocardial fibrosis and remodeling in heart failure by the concentration of urine IL-4 in patients with HF and its relationship to markers of myocardial fibrosis and left ventricular volume [80].

IL-4 mediates the upregulation of proto-collagen genes through IL-4 receptor α and stimulates collagen production in mouse cardiac fibroblasts. This sheds light on the critical role of IL-4 in angiotensin-II-induced cardiac injury and provides a strong basis for IL-4 as an additional target for the treatment of cardiac fibrosis [81].

### 3.3. IL-6

IL-6 is another proinflammatory cytokine that has been implicated in the development of fibrosis. In an array of heart diseases, IL-6 promotes fibroblast proliferation and stimulates ECM synthesis [82,83,84].

Silencing IL-6 in EDC eliminated most of the benefits of cell transplantation and showed that IL-6 promotes cardiac fibroblasts and macrophages to reduce myocardial scarring while increasing the production of new cardiomyocytes and the recruitment of hematopoietic stem cells. IL-6 plays a key role in EDC-mediated cardiac repair and may provide a way to increase cell-mediated ischemic myocardial repair [85].

Both pro- and anti-inflammatory effects of IL-6 have been reported. In the short term, IL-6 has a protective effect and limits host damage, and it is precisely when this acute response remains chronically activated that IL-6 becomes pathogenic to the host; long-term elevated IL-6 levels lead to chronic inflammation and fibrotic disease. It is supposed that short-term IL-6 signaling protects and preserves heart tissue in response to acute injury; moreover, long-term IL-6 signaling or overproduction of the IL-6R protein plays a causal role in cardiovascular disease. Depending on the kinetics of the host response, IL-6 can be both protective and pathogenic [86].

### 3.4. IL-10

IL-10 is an anti-inflammatory cytokine that can have protective effects on the heart. IL-10 knockout mice presented worse left ventricular function and fibrosis following MI [87]. The IL-10 treatment demonstrated the downregulation of p38 mitogen-activated protein kinase activation, reduced expression of the cytokine mRNA-stabilizing protein Hur, a decreased metalloproteinase-9 (MMP-9) activity, and inhibited fibrosis after MI [88]. IL-10 gene expression inhibition is associated with the suppression of TLR4 and IL-1 receptor-associated kinase-1 (IRAK1) activation along with the upregulation of TLR2 and IRAK2, resulting in fibrosis.

### 3.5. IL-11

It is reported that IL-11 plays a profibrotic effect as a downstream effector following TGFβ1 exposure; the genetic deletion of IL-11 protects the heart from fibrosis [4]. In contrast to this, IL-11 attenuated cardiac fibrosis and remodeling after MI through the gp130/STAT3 axis [5]. Therefore, the exact role of IL-11 in heart fibrosis needs to be further elucidated.

### 3.6. IL-17

Th17 cells produce MMP through the IL-17-RANKL/OPG system in cardiac fibroblasts, or regulate cardiac fibrosis by stabilizing mRNAs from proinflammatory cytokines in various cardiomyocytes and immune cells [89]. IL-17 has been shown to exaggerate the extent of cardiac fibrosis, the activation of MMPs, and enhanced cardiac cell death.

### 3.7. IL-18

IL-18 and osteopontin (OPN) gene and protein expression were found elevated in pressure-overload mice; IL-18 has an effect on mice cardiac fibroblasts to induce OPN [90]. In addition to this, IL-18 has profibrotic effects on human cardiac fibroblasts by inducing fibronectin production [91]. Furthermore, IL-18 has profibrotic effects on rat cardiac fibroblasts by producing collagen type I and III, MMP-2, and then activating the JNK and PI3-kinase pathways. In addition to this, IL-18 leads to fibroblast migration and proliferation. These studies illustrate that IL-18 plays an important role in cardiac fibrosis-related disorders [92].

### 3.8. IL-33

As we mentioned before, recombinant IL-33 treatment reduced angiotensin II- and phenylephrine-induced fibrosis and improved survival after TAC in WT mice [72].

### 3.9. IL-37

The Zeng group reported that the intraperitoneal injection of human recombinant IL-37 and the intravenous injection of IL-37 and troponin co-induced dendritic cells can alleviate the adverse ventricular remodeling after MI and attenuated the degree of cardiac fibrosis in mice [77].

Overall, the effects of interleukins on fibroblasts in heart disease are complex and context-dependent, and more research is needed to fully understand their roles in the development of fibrosis and other pathological processes in the heart.

## 4. Endothelial Cells (ECs)

ECs are regarded as the major non-CM population in the heart, suggesting that their physiological and therapeutic importance may be greater than previously appreciated. ECs comprise approximately 95% of the blood vascular and 5% of the lymphatic system and are responsible for forming blood vessels and valves.

Furthermore, ECs play an important role in angiogenesis, which is the formation of new blood vessels from pre-existing ones. ECs also participate in the development of the heart and lymphatic system during embryonic development. The specific effects of different interleukins on ECs in common heart disease are summarized in Figure 3.

### 4.1. IL-1

IL-1α has been shown to stimulate the expression of macrophage colony-stimulating factor (MCSF) in both ECs and SMCs at both the mRNA and protein levels during atherogenesis. This upregulation of MCSF may play a critical role in the formation of atherosclerotic lesions and the progression of MI. ECs and SMCs are known to contribute to the development of atherosclerosis and MI, and the increased expression of MCSF may promote the recruitment of monocytes and macrophages to the site of injury, leading to the formation of atherosclerotic plaques. These findings suggest that targeting the IL-1α-MCSF signaling pathway may hold therapeutic potential for the prevention and treatment of atherosclerosis and MI. Further studies are needed to fully elucidate the mechanisms underlying this pathway and to identify potential therapeutic targets [93].

### 4.2. IL-2

In a recent study, it was found that the injection of recombinant human interleukin-2 (rhIL-2) into mice led to a significant increase in the proliferation index of endothelial cells in the infarcted heart following MI. Specifically, the EC proliferation index was enhanced by 1.6-fold compared to control mice, indicating a potential role for rhIL-2 in promoting cardiac tissue repair following MI. These findings suggest that rhIL-2 may hold therapeutic potential for the treatment of MI and other cardiovascular diseases by promoting EC proliferation and angiogenesis. Further studies are needed to fully elucidate the underlying mechanisms and to optimize the dosing and delivery of rhIL-2 for clinical use [29].

### 4.3. IL-8

IL-8 may play a protective role in angiogenesis function; Haleagrahara et al. reported that in rats receiving insulin-like growth factor 1 (IGF-1) treatment, the enhanced angiogenetic impact of IL-8 is linked to the protection of ischemic myocardium and isoproterenol-induced cardiac damage [94]. In AMIs, IL-8 is associated with circulating progenitor cells, and in addition to the proangiogenesis function of IL-8 and VEGF, this mechanism may contribute to the production of new blood vessels and ECs, thereby improving myocardial function [95]. Therefore, the therapeutic potential in heart disease of IL-8 should be further developed in the future.

### 4.4. IL-10

IL-10 deficiency and inflammation have been shown to affect the function and content of exosomes derived from endothelial progenitor cells (EPCs), which in turn affects their therapeutic efficacy in myocardial repair. This effect is mediated by an upregulation of integrin-linked kinase (ILK) enrichment in exosomes, which activates the NF-κB pathway in recipient cells. Conversely, the knockdown of ILK in exosomes attenuates NF-κB activation and reduces the inflammatory response. These findings suggest that ILK may play a crucial role in modulating the therapeutic potential of EPC-derived exosomes in the context of myocardial repair [96]. Further research is needed to elucidate the underlying mechanisms and potential clinical applications of these findings.

### 4.5. IL-17

IL-17 has been shown to increase nitric oxide (NO) synthesis, leading to endothelial cell (EC) injury in an oxygen-dependent manner [97]. Additionally, IL-17 has been found to induce the expression of adhesion molecules, including CXCL1 (GRO-α) and CXCL8 in ECs [98]. These findings suggest that IL-17 may play a role in the development of vascular inflammation and injury, which are key components of many cardiovascular diseases. Further research is needed to fully elucidate the mechanisms underlying the effects of IL-17 on EC function and to explore potential therapeutic targets for the treatment of cardiovascular disease.

### 4.6. IL-18

In patients with MI who undergo coronary artery bypass grafting (CABG), elevated levels of IL-18 in the systemic circulation have been reported to activate lymphocytes, which may subsequently lead to EC cytotoxicity [99]. I/R injury triggers the activation of the NLRP3/IL-18 signaling pathway, which in turn induces the transcription of CXCL16 in cardiac vascular endothelial cells (VECs). Furthermore, it was revealed that the transcription of CXCL16 is dependent on the transcription factor FOXO3 and that IL-18-mediated STAT3 phosphorylation promotes nuclear translocation of FOXO3, which enhances CXCL16 promoter activity by binding to the FOXO3 binding site. These findings suggest that CXCL16 may serve as a potential therapeutic target for treating IL-18-mediated myocardial I/R injury by reducing cardiac inflammation and improving cardiac remodeling and dysfunction [100]. It was suggested that IL-18 may play a significant role in the pathogenesis of cardiovascular disease by promoting inflammation and vascular injury.

### 4.7. IL-33

IL-33 is a cytokine that can promote angiogenesis and vascular leakage. Experimental studies have shown that IL-33 stimulates ECs to produce nitric oxide (NO) through the ST2/TRAF6-Akt-eNOS signaling pathway, thereby promoting the process of angiogenesis and vascular leakage. This discovery reveals the mechanism of action of IL-33 in vascular diseases and provides new ideas for the development of treatments for related diseases. This study provides new evidence for revealing the mechanism of the IL-33 signaling pathway in angiogenesis and vascular leakage and provides new ideas and strategies for the treatment of related diseases [101].

Overall, the effects of interleukins on ECs in heart disease are mostly on EC proliferation and angiogenesis. More research is needed to fully understand other interleukins in the development of angiogenesis and other pathological processes in the heart.

## 5. Immune Cells

Immune cells are implicated in the pathophysiology of heart diseases and include several types of cells in the heart, such as T cells, B cells, macrophages, neutrophils, and NK cells. The specific effects of different interleukins on these immune cells in common heart disease are summarized in Figure 4.

### 5.1. IL-1

IL-1 critically participates in the process of post-MI by activating leukocytes and fibroblasts, and IL-1 signals the absence of globally attenuated leukocyte recruitment, reducing the number of infiltrating Ly6C(hi) and Ly6C(lo) cells [78]. IL-1α was suggested to be a mediator of M2-like macrophage-induced fibroblast activation [102]. In acute MI, targeting IL-1β reduces leukocyte production and inflammation by suppressing bone marrow hematopoietic stem cell proliferation, demonstrating the positive influence of IL-1β on leukocyte production [103]. Furthermore, IL-1 receptor type 1 (IL-1R1) is critical for the induction of autoimmune myocarditis; it is required for dendritic cells producing TNF-α, IL-1, IL-6, and IL-12p70. In addition, the triggering of IL-1R1 induces CD4(+) T-cell activation and autoimmunity [104].

### 5.2. IL-2

It is confirmed that a high level of IL-2 can induce T-cell growth and differentiation and enhances NK cell activation and mediated cell death [7]. In DCM patients, elevated IL-2 levels expressed by CD4+ T lymphocytes were observed, then the cytokine stimulates macrophages to regulate the innate immune process [105]. In contrast, IL-2 plus IL-2 monoclonal antibody clone JES6-1 complexes (IL2/JES6-1) administration increased Tregs, suppressed leukocyte infiltration, including decreasing CD45(+) cells, macrophages, CD8(+) T cells, and effector memory CD8(+) in mice hearts before TAC-induced CHF, then attenuated the development of LV hypertrophy and dysfunction [106]. The injection of IL-2-activated NK cells promotes vascular remodeling and promotes cardiac repair after MI, as mentioned earlier [29].

### 5.3. IL-4

Recombinant protein IL-4 was used as a drug for treating MI and may activate M2 macrophage accumulation to enhance cardiac function, reduce infarction size, and repair damaged tissues [8]. Another study demonstrated that microRNA-155(−/−) mice developed attenuated viral myocarditis because of expressing increased levels of IL-4, affecting macrophage polarization to confer potential therapeutic targets for viral myocarditis [33].

### 5.4. IL-6

Systemically high levels of IL-6 produced by PBMC in HF patients affect NK cells through signal transduction pathways and may also affect other cells involved in heart disease, including cardiomyocytes, and possibly the function of other immune cells. In the model of LV remodeling after MI, a drug blockade of IL-6, through the administration of the anti-IL6R antibody, decreases neutrophil and macrophage infiltration in the infarct region, and the administration of anti-IL6R antibody after MI suppressed myocardial inflammation resulted in the amelioration of LV remodeling [41]. Another study has demonstrated that an IL-6 receptor antagonist can effectively attenuate the inflammatory response in patients with non-ST-elevation myocardial infarction (NSTEMI) [37]. In conclusion, all of these results show the harmful effect of IL-6 on the immune system in heart disease.

### 5.5. IL-8

IL-8 has been identified as a chemokine that plays a critical role in mediating neutrophil activation and recruitment, as well as promoting angiogenesis in healing infarcts [107]. IL-8 was seen to be associated with increasing circulating CD133+ progenitor cells in AMI patients, which then leads to new vessel generation and improved MI function [108]. However, recombinant canine IL-8 markedly increased the adhesion of neutrophils to CMs resulting in cytotoxicity in I/R canine myocardium [109].

### 5.6. IL-10

Anti-IL-10 therapy canceled the protective effect mediated by H310A1 infection by inhibiting T regulatory cell response, indicating that IL-10 is the main immunomodulatory factor in viral myocarditis [110].

### 5.7. IL-17

IL-17 plays an important role in the inflammation phase of heart disease. It has biological effects on neutrophil recruitment, leukocytes, and monocyte infiltration; however, there is also evidence of its activation of M2 macrophages for restoring cardiac damage.

IL-17A knockout mice significantly attenuated cardiomyocyte apoptosis and neutrophil infiltration in myocardial I/R injury, demonstrating the primary role of IL-17 in neutrophil recruitment [111]. In addition to this, IL-17 induces the accumulation of leukocytes in an inflammatory state, such as AMI, because of the adhesion molecules effect [112]. IL-17 is essential for the infiltration of Ly6C hi monocytes into inflammation sites of infarcted myocardial tissue [113]. This evidence suggests the deleterious property of IL-17 on the inflammation phase of heart disease.

On the other hand, it induces the production of GM-CSF, which could contribute to the differentiation of M1 macrophages. It also participates in the activation of M2 macrophages, which promote inflammatory healing, angiogenesis, and tissue remodeling [114]. Therefore, IL-17 plays a dual role in the inflammatory response of heart disease.

### 5.8. IL-18

The role of IL-18 in different types of heart disease is not consistent and is even controversial. For instance, IL-18 reduces the severity of viral myocarditis by inducing the cardiac expression of IFN-γ mRNA and increasing the activity of natural killer cells in the spleen [115]. In addition, in MI patients undergoing coronary artery bypass grafting (CABG), IL-18 in systemic circulation leads to the activation of lymphocyte cytotoxicity, then enhancing circulating lymphocyte activity [99].

### 5.9. IL-33

IL-33 knockout mice showed elevated Th1 cytokine expression levels and the infiltration of inflammatory cells in the myocardium of HF induced by mechanical stress [116]. In contrast to this finding, IL-33-treated mice displayed eosinophil recruitment and worsened systolic dysfunction at 7 days post-MI [117].

### 5.10. IL-37

As an innate immune anti-inflammatory interleukin, human recombinant IL-37 inhibits neutrophil infiltration through tail vein injection into myocardial I/R mice [76]. IL-37 has a protective effect on myocardial I/R injury by promoting Treg cell activation [118]. Overall, the therapeutic potential of IL-37 needs to be further developed and investigated.

Therefore, it is important to note that the effects of interleukins on immune cells in heart disease can vary depending on the specific type of heart disease and the stage of the disease. Further research is needed to fully understand the complex interactions between immune cells and interleukins in the pathophysiology of heart disease.

The various deleterious, protective, and dual roles of interleukin family members on primary cell types of heart disease are summarized in Figure 5.

## 6. Conclusions

The inflammation process might be responsible for the initiation and progression of heart disease. Different IL families have been shown to be associated with the development of heart disease. The use of proinflammatory factors such as IL-1β antagonists or antibodies has made positive progress in heart disease early clinical trials and researchers continue to explore more effective cytokine treatment methods. However, the exact role of interleukins and their pathophysiological pathways on different cell types remain largely unknown. Additionally, several interleukins can act as both pro- and anti-inflammatory factors, and have either cardioprotective or deleterious properties, depending on the cell type.

In this review, the pathophysiological effects of interleukins including the IL-1 family (IL-1, IL-18, IL-33, and IL-37), IL-2, IL-4, the IL-6 family (IL-6 and IL-11), IL-8, IL-10, and IL-17 on primary cell types of heart disease is elucidated (Figure 1). It is concluded that determining the exact role of any particular cytokine in the pathogenesis and progression of different cell types in the heart is of significance. Targeting a specific cell type for the treatment of heart disease is likely to be more effective, and further research into IL therapy for specific cell types is needed to improve the quality of life and survival rate of patients.

## Figures and Tables

**Figure 1 ijms-24-06497-f001:**
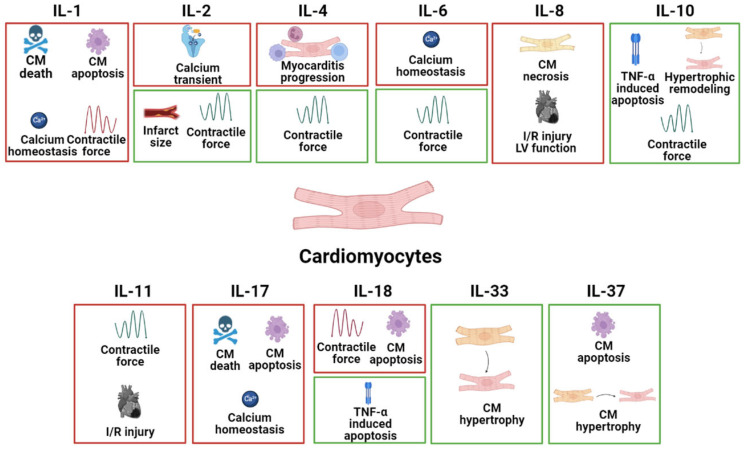
The specific pathophysiological effects of different interleukins on CMs in common heart disease. The box in red represents the deleterious role, and the box in green represents the protective role.

**Figure 2 ijms-24-06497-f002:**
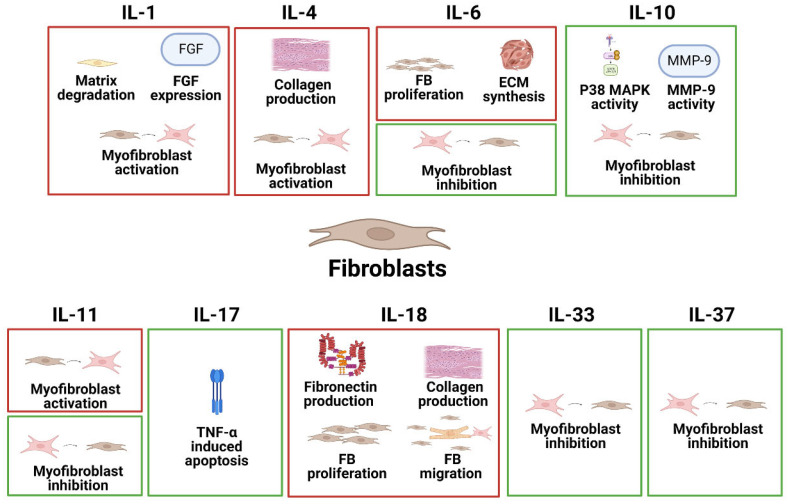
The specific pathophysiological effects of different interleukins on FBs in common heart disease. The box in red represents the deleterious role, and the box in green represents the protective role.

**Figure 3 ijms-24-06497-f003:**
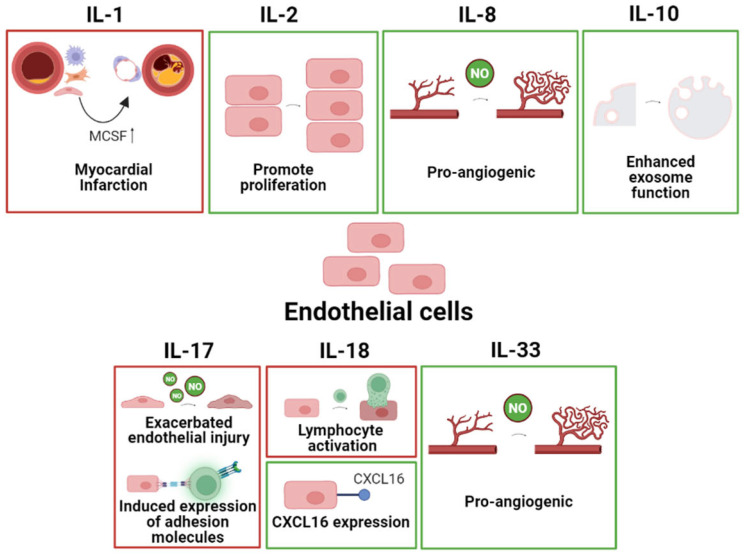
The specific pathophysiological effects of different interleukins on ECs in common heart disease. The box in red represents the deleterious role, and the box in green represents the protective role.

**Figure 4 ijms-24-06497-f004:**
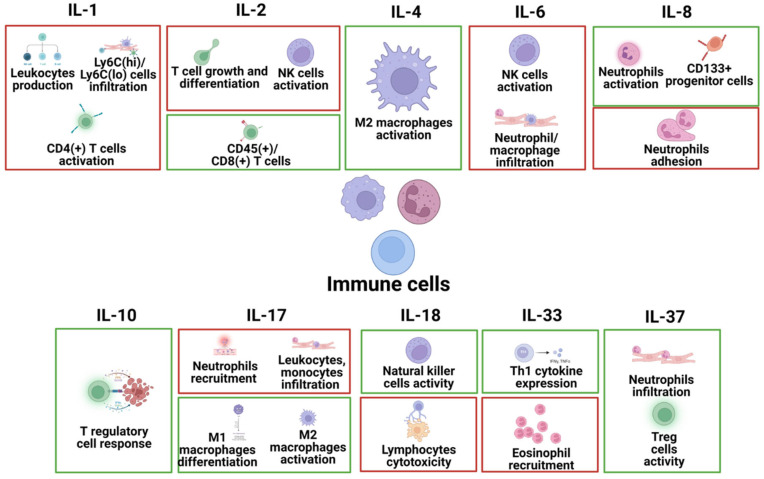
The specific pathophysiological effects of different interleukins on immune cells in common heart disease. The box in red represents the deleterious role, and the box in green represents the protective role.

**Figure 5 ijms-24-06497-f005:**
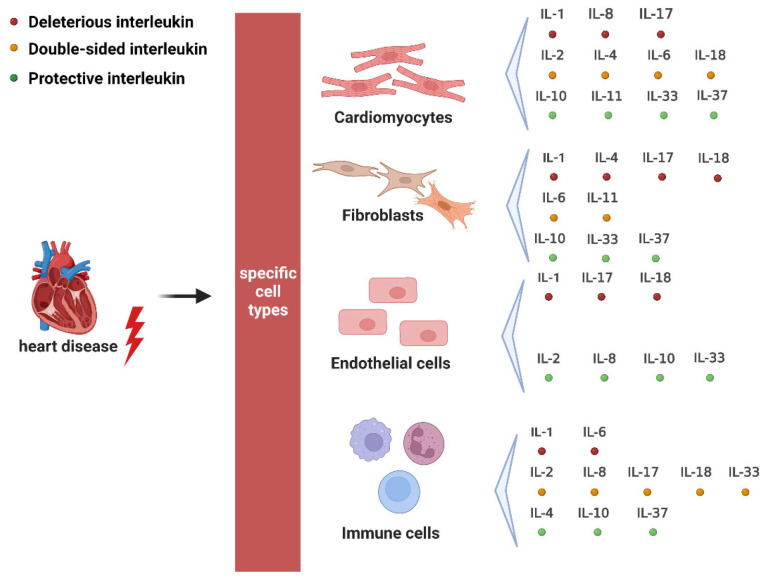
The overall effects of interleukin family members on primary cell types in heart disease. The heart comprises these major cell types: cardiomyocytes (CMs), fibroblasts (FBs), endothelial cells (ECs), and immune cells. An interleukin marked with a red circle represents the deleterious effect on a specific cell type, a green circle represents the protective effect on a specific cell type, and an orange circle represents a double-sided effect.

## Data Availability

Not applicable.

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
