# Peer review of "Pathophysiological Effects of Various Interleukins on Primary Cell Types in Common Heart Disease"

_ijms, 2023, doi:10.3390/ijms24076497_

Round 1
Reviewer 1 Report
The paper entitled "Pathophysiological role of interleukins on primary cell types in heart disease" is dealing with interesting and complex topic of interleukins interplay in heart disease. The authors have sistematically presented pletohra of data related to the topic. However there are some issues to be resolved before reconsidering for publication.
1. English language was ocasionally very difficult to understand - e.g. lines 88-89; 191-196; 219-222; 225-226; 412-413; 416-417...
2. Introduction section - after line 38 text is missing - the paragraph ends abruptly
3. lines 52-58 - references should be provided in 2 paragraphs dealing with IL-2 and IL-4
4. Data on IL affecting endothelial cells should be elaborated. In some cases there is just a single sentence describing the effect of IL on endothelial cells.
5. Additional paragraph in conclusion section (or at the end of every section related to different cell types) would be of benefit. Although there is lots of controversy(opposite effects sometimes) regarding the overall effect of ILs in different cell types, it would be beneficial if authors would sum up the presented data. It would increase the quality of the paper if there is an author opnion/point of view presented instead of just listing/citing data.
Author Response
Dear Reviewer:
Thank you for your comments concerning our manuscript entitled “Pathophysiological role of interleukins on primary cell types in heart disease” (ID: ijms-2263501). These comments are all valuable and very helpful for revising and improving our paper. We have studied comments carefully and have made correction which we hope meet with approval. All revisions to the manuscript have been marked up using the “Track Changes” function in MS Word. The main corrections in the paper and the responds to the reviewer’s comments are as following:
Point 1: English language was occasionally very difficult to understand - e.g. lines 88-89; 191-196; 219-222; 225-226; 412-413; 416-417.
Response 1: We have re-written these parts according to the Reviewer’s suggestions.
Point 2: Introduction section - after line 38 text is missing - the paragraph ends abruptly.
Response 2: We have made corrections according to the Reviewer’s comments.
Point 3: lines 52-58 - references should be provided in 2 paragraphs dealing with IL-2 and IL-4.
Response 3: We have added references for these two paragraphs.
Point 4: Data on IL affecting endothelial cells should be elaborated. In some cases there is just a single sentence describing the effect of IL on endothelial cells.
Response 4: We have made modifications involved in the specific effects of different interleukins on ECs in common heart disease, and have added a figure for ECs.
Point 5: Additional paragraph in conclusion section (or at the end of every section related to different cell types) would be of benefit.
Response 5: We have concluded additional paragraph at the end of each section.
We tried our best to improve the manuscript and made some changes in the manuscript. These changes will not influence the content and framework of the paper. We appreciate for your warm work earnestly, and hope that the correction will meet with approval. Once again, thank you very much for your comments and suggestions.
Reviewer 2 Report
The present manuscript discussed the functions of interleukins on different cardiac cell types in pathophysiological conditions. Overall, the manuscript of this review article is comprehensive and brings some novelty in the research field. However, there are some MAJOR concerns that need to be clarified. Besides, there are many technical issues regarding this manuscript. Therefore, the whole manuscript needs to be thoroughly checked and modified.
First, the research methodology should be more in-depth explained. What was the literature search strategy, which databases were used, how many articles were included, were there some papers that were excluded and why, etc?
Given that the title refers to ‘primary cells’, it should be clarified that there are both resident and recruited (infiltrating) immune cells that play a role in cardiac injury and that are present in cardiac tissue early in disease, i.e. to emphasize that some of the effects are the consequence of the heart immune cells infiltration during inflammation.
Both in Introduction and Conclusions, the authors state that proinflammatory factors such as IL-1β 'have shown promising results in early clinical trials for heart diseases'. However, there are no references for that and the truth is actually the opposite - suppression of IL-1β results in a prompt and persistent decrease in disease severity. Later in the text (line 105), the authors state that ‘IL-1α blockers may represent an effective therapeutic approach to reduce I/R damage to the heart’, which is correct.
The figures in the text are illustrative and give the overview of the text. Why isn’t there a figure for endothelial cells? It should be added. Besides, in my opinion figure 1 should be at the end of the manuscript since it summarizes the whole manuscript. In Introduction, the emphasis should be on different cell types in heart diseases and generally on cytokines in the disease development.
Some other concerns include:
Please format the text according to the journal instructions – for example lines 32-39 and lines 320-327 (it shouldn’t be bold).
Line 25 – add the reference for this epidemiological data.
Line 26 – failure instead of Failure
Line 62 – neuropoietin instead of Neuropoietin
Line 83 (IL plays a key role in the pathogenesis of heart disease) – we are not completely sure if they have the ‘key’ role, but ‘significant’ definitely yes.
In Figures – homeostasis instead of homeostas
Line 231 – the instead of The
Please define TAC and WT in the text.
Line 337 – what does ‘(6)’ refer to?
Lines 359-360 – why are some parts underlined and hyperlinked?
Line 373 – immune instead of Immune
There are many other technical issues in this manuscript and the whole manuscript, thus, needs to be thoroughly checked.
Author Response
Dear Reviewer:
Thank you for your comments concerning our manuscript entitled “Pathophysiological role of interleukins on primary cell types in heart disease” (ID: ijms-2263501). These comments are all valuable and very helpful for revising and improving our paper. We have studied comments carefully and have made correction which we hope meet with approval. All revisions to the manuscript have been marked up using the “Track Changes” function in MS Word. The main corrections in the paper and the responds to the reviewer’s comments are as following:
Point 1: First, the research methodology should be more in-depth explained. What was the literature search strategy, which databases were used, how many articles were included, were there some papers that were excluded and why, etc.
Response 1: We have added related sentences in the manuscript: We have searched papers by using Pubmed and Google scholar with keywords: interleukins, IL-1, IL-18, IL-33, IL-37, IL-2, IL-4, IL-6, IL-11, IL-8, IL-10, IL-17 and heart disease. Then we have selected approximately 120 papers involved pathophysiological role of interleukins on primary cell types in heart disease as reference.
Point 2: Given that the title refers to ‘primary cells’, it should be clarified that there are both resident and recruited (infiltrating) immune cells that play a role in cardiac injury and that are present in cardiac tissue early in disease, i.e. to emphasize that some of the effects are the consequence of the heart immune cells infiltration during inflammation.
Response 2: We have added related sentences in the manuscript: CMs, FBs, ECs are resident cells in the heart, however immunes cells are infiltrated during inflammation, both of them play significant role in the pathophysiological process of heart disease.
Point 3: Both in Introduction and Conclusions, the authors state that proinflammatory factors such as IL-1β 'have shown promising results in early clinical trials for heart diseases'. However, there are no references for that and the truth is actually the opposite - suppression of IL-1β results in a prompt and persistent decrease in disease severity. Later in the text (line 105), the authors state that ‘IL-1α blockers may represent an effective therapeutic approach to reduce I/R damage to the heart’, which is correct.
Response 3: Sorry for the inappropriate statement, we would like to state: Proinflammatory factor IL-1β antagonist or antibody have shown promising results in early clinical trials for heart disease. We have added reference for this sentence in the manuscript.
Point 4: The figures in the text are illustrative and give the overview of the text. Why isn’t there a figure for endothelial cells? It should be added. Besides, in my opinion figure 1 should be at the end of the manuscript since it summarizes the whole manuscript. In Introduction, the emphasis should be on different cell types in heart diseases and generally on cytokines in the disease development.
Response 4: We have added a figure for endothelial cells. Figure 1 has been moved at the end of the manuscript.
Other changes:
- Lines 32-39, lines 320-327, Line 25, Line 26, Line 62, Line 83, Line 231, Line 337, Lines 359-360, Line 373. We have made correction according to the Reviewer’s comments.
We tried our best to improve the manuscript and made some changes in the manuscript. These changes will not influence the content and framework of the paper. We appreciate for your warm work earnestly, and hope that the correction will meet with approval. Once again, thank you very much for your comments and suggestions.
Round 2
Reviewer 1 Report
The authors have improved the manuscript according to suggestions given in review round 1. I suggest for the reviewed version to be published in present form.
Reviewer 2 Report
The manuscript is considerably improved and I support it to be published in this form.